# Regulation of the Actin Cytoskeleton-Linked Ca^2+^ Signaling by Intracellular pH in Fertilized Eggs of Sea Urchin

**DOI:** 10.3390/cells11091496

**Published:** 2022-04-29

**Authors:** Nunzia Limatola, Jong Tai Chun, Luigia Santella

**Affiliations:** 1Department of Research Infrastructures for Marine Biological Resources, Stazione Zoologica Anton Dohrn, 80121 Napoli, Italy; 2Department of Biology and Evolution of Marine Organisms, Stazione Zoologica Anton Dohrn, 80121 Napoli, Italy; chun@szn.it

**Keywords:** fertilization, sea urchin eggs, polyspermy, actin, Ca^2+^ signaling, cortical reaction, intracellular pH, weak bases

## Abstract

In sea urchin, the immediate contact of the acrosome-reacted sperm with the egg surface triggers a series of structural and ionic changes in the egg cortex. Within one minute after sperm fuses with the egg plasma membrane, the cell membrane potential changes with the concurrent increases in intracellular Ca^2+^ levels. The consequent exocytosis of the cortical granules induces separation of the vitelline layer from the egg plasma membrane. While these cortical changes are presumed to prevent the fusion of additional sperm, the subsequent late phase (between 1 and 4 min after fertilization) is characterized by reorganization of the egg cortex and microvilli (elongation) and by the metabolic shift to activate de novo protein and DNA syntheses. The latter biosynthetic events are crucial for embryonic development. Previous studies suggested that the early phase of fertilization was not a prerequisite for these changes in the second phase since the increase in the intracellular pH induced by the exposure of unfertilized sea urchin eggs to ammonia seawater could start metabolic egg activation in the absence of the cortical granule exocytosis. In the present study, we have demonstrated that the incubation of unfertilized eggs in ammonia seawater induced considerable elongations of microvilli (containing actin filaments) as a consequence of the intracellular pH increase, which increased the egg’s receptivity to sperm and made the eggs polyspermic at fertilization despite the elevation of the fertilization envelope (FE). These eggs also displayed compromised Ca^2+^ signals at fertilization, as the amplitude of the cortical flash was significantly reduced and the elevated intracellular Ca^2+^ level declined much faster. These results have also highlighted the importance of the increased internal pH in regulating Ca^2+^ signaling and the microvillar actin cytoskeleton during the late phase of the fertilization process.

## 1. Introduction

The fertilization of echinoderm eggs (starfish and sea urchin) in seawater comprises early structural and ionic changes in the gametes. When sperm contacts the jelly coat (JC) of the egg, the components of the JC induce exocytosis of the acrosome vesicle in the sperm head, and the concurrent polymerization of F-actin promotes the formation of the acrosomal process [1]. In starfish, it is possible to induce in vitro the formation of the long and thin acrosomal filament (approximately 20 µm in length) [2,3]. This morphological feature has also allowed for the recording of the electrical response of the eggs at fertilization upon the contact of the tip of the sperm acrosomal process with the egg plasma membrane and for the measurement of the Ca^2+^ signals before and during the separation of the vitelline layer (VL) from the plasma membrane [4,5,6,7]. In sea urchin sperm, bindin, an adhesive protein that is secreted and exposed on the surface of the acrosomal process, has been suggested to be responsible for the species-specific binding of the sperm to the glycoprotein receptors on the VL covering the egg plasma membrane [8,9,10,11]. Upon the fusion of the sperm with the egg plasma membrane, electrophysiological measurements of the eggs of several sea urchin species have detected an initial response of the egg in the form of an increase in the plasma membrane conductance (plasma membrane depolarization) due to a flow of Ca^2+^ and Na^+^ into the egg, which is followed by a fertilization potential [12,13]. The existence of a fast mechanism to allow for the fusion of only one sperm was suggested more than one hundred years ago [14,15], and the prevailing idea is that the fast membrane depolarization (1 to 3 s after insemination) could electrically block the entry of multiple sperm [16]. This motion has also been challenged recently with another sea urchin species (*Paracentrotus lividus*), in which eggs fertilized in artificial seawater containing low Na^+^ remained monospermic. The observed abnormal embryonic development of these zygotes resulted from the altered F-actin structure and dynamics in the eggs fertilized in low Na^+^ media and not from the formation of multiple polar spindles related to polyspermy [17]. The results also showed that a structural block to polyspermy is established by the attachment of the fertilizing sperm to the egg surface, which prevents the binding and fusion of additional sperm before the separation of the vitelline layer [14,17].

Ca^2+^ imaging of the fertilization response with temporal and spatial resolution performed in intact starfish and sea urchin eggs has revealed that the electrophysiological response induced by the fertilizing sperm precisely mirrors the intracellular calcium response [4,18]. In *P. lividus* eggs, the first step of depolarization (the so-called “latent period”), recorded a few sec after insemination [19], is concurrent with the simultaneous increase in Ca^2+^ in the egg cortex (cortical flash, CF) due to the influx through the L-type calcium channels [20,21,22]. The latent period is followed by the fertilization potential, which corresponds to the intracellular Ca^2+^ wave, propagating from the site of sperm–egg fusion to the opposite pole [18].

Identification of the signal transduction pathways leading to the intracellular Ca^2+^ changes experienced by the fertilized eggs has been the subject of investigation for decades. Results from experiments conducted in the 1980s on sea urchin and starfish eggs suggested that the fertilizing sperm triggers the activation of phospholipase C isoforms through heterotrimeric G proteins or tyrosine kinases (Src family kinases) to produce inositol 1,4,5-trisphosphate (IP_3_), which releases Ca^2+^ from its receptor ion channels on the endoplasmic reticulum [23,24,25]. Alternative hypotheses of egg activation also suggested that the sperm introduced a bolus of Ca^2+^ or a soluble factor into the egg during the gamete fusion to initiate the Ca^2+^ release [26,27]. Later studies pointed to the role of the nicotinic acid adenine dinucleotide phosphate (NAADP) signaling pathway in initiating the sperm-induced Ca^2+^ response of starfish and sea urchin eggs. Still, they suggested that NAADP might have different modes of action in the eggs of these two echinoderms. Whereas NAADP targets distinct Ca^2+^ stores such as acidic vesicles in the cortex of sea urchin eggs, NAADP in starfish eggs induces membrane depolarization and Ca^2+^ influx [28,29,30,31,32,33]. One of the primary roles of the Ca^2+^ signals at fertilization is the activation of NAD kinase, which contributes to the generation of H_2_O_2_ in collaboration with NADP/H oxidase in the plasma membrane to crosslink the FE [34,35].

The outmost cortex of unfertilized eggs of starfish and sea urchin typically exhibits short microvilli and actin filaments to which cortical granules are tightly associated. Recent findings have revealed that the F-actin-dependent structural organization of the egg cortex upon insemination plays a crucial role in controlling the temporal and spatial phases of the fertilization response in starfish and sea urchin eggs [7,18,36,37,38,39]. In line with this, the structural alteration of microvilli and cortical granules in the cortex of *P. lividus* eggs is linked to compromised egg activation following fertilization, e.g., anomalies in the generation of Ca^2+^ signals and the separation of the VL [17,22,40,41,42]. The cytoskeletal modifications of the egg cortex occurring 5 min after insemination and during the decline of the intracellular Ca^2+^ increase to the baseline include microvillar elongation in the perivitelline space and the lifting of the fertilization envelope (FE) [5,43,44,45]. These cortical changes require fine regulation of actin dynamics. Indeed, the functional integrity of the actin cytoskeleton is an index of the egg quality, ensuring physiological responses to the fertilizing sperm.

In sea urchin eggs, what takes place concomitantly with the structural changes in the egg cortex during the late phase of fertilization (5 min after insemination) includes: (i) an increase in intracellular pH due to protons efflux (fertilization acid) [46,47], (ii) the development of new K^+^ conductance [48] and metabolic derepressions that are crucial for the embryonic development, i.e., protein synthesis and chromosome condensation [49,50], and the polyadenylation of cytoplasmic messenger RNA [51].

The findings that the late metabolic activation seen in the fertilized sea urchin eggs could be triggered in unfertilized eggs using membrane-permeant weak bases such as ammonium hydroxide (NH_4_OH), ammonium chloride (NH_4_Cl), nicotine, and procaine [52,53] in the absence of cortical granule exocytosis [54] led to the suggestion that the increase in intracellular pH might be sufficient for cell division and differentiation [48]. These results weakened the claimed critical roles played by plasma membrane depolarization and Ca^2+^-linked breakdown of the cortical granules in inducing metabolic derepression. However, later studies on sea urchin egg cortices showed that not only the activation of biosynthetic events but also the extension of microvilli based on actin polymerization require an increase in both intracellular pH and Ca^2+^ [55,56,57].

The present study has examined the effect of the seawater titrated to pH 9 by NH_4_OH on sea urchin eggs concerning the surface topography and the fertilization reaction. The results have indicated that ammonia seawater induced dramatic alteration of the structural organization of the egg surface as a result of the alkalinization of the egg cytoplasm [53]. The cortical structural changes included microvilli elongation, which increased the receptivity of the eggs to multiple sperm. Upon fusing with the eggs in ammonia seawater, multiple sperm could enter the egg readily despite the elevation of the FE. Microvilli elongation before fertilization and overextension after insemination also compromised the sperm-induced Ca^2+^ signals, i.e., it reduced the amplitude of the initial CF and accelerated the falling phase of the Ca^2+^ level to the baseline during the late phase. The results have highlighted the strict relationship between the increase in internal pH and the modulation of the intracellular Ca^2+^ levels mediated by actin polymerization in microvilli.

## 2. Materials and Methods

### 2.1. Gametes Collection, Modification of Seawater pH, and Embryos Observation

During the breeding season, adult specimens of *P. lividus* (October to May) and *Arbacia lixula* (May to September) were collected in the Gulf of Naples and maintained at 16 °C in circulating seawater. Artificial spawning was induced by the intracoelomic injection of 0.5 M KCl. The released eggs were collected in natural seawater (NSW) filtered with a Millipore membrane of 0.2 μm pore size (Nalgene vacuum filtration system, Rochester, NY, USA) and used for experiments shortly thereafter. For fertilization, “dry sperm” were collected by pipetting on the male animal’s body surface and kept at 4 °C. A few minutes before fertilization, dry sperm were diluted in NSW at a final concentration of 1.84 × 10^6^ units/mL. In this work, the incubation medium referred to as “ammonia seawater” was obtained by adding NH_4_OH to NSW until reaching pH 9 (about 1 mM NH_4_OH). In most experiments, eggs suspended in NSW (pH 8.1) were transferred to ammonia seawater at pH 9 and incubated for 20 min before fertilization. In three independent experiments, the embryonic development of 100 eggs from different batches incubated and inseminated in ammonia seawater was observed with a Leica DMI6000 B inverted microscope (Leica Microsystems, Wetzlar, Germany).

### 2.2. Visualization of Egg-Incorporated Sperm

Sperm were stained with 5 µM Hoechst-33342 (Sigma–Aldrich, Saint Louis, MO, USA) for 30 s before insemination. The labeled sperm nuclei incorporated into *P. lividus* eggs were visualized and counted 5 min after insemination using a cooled CCD (charge-coupled device) camera (MicroMax, Princeton Instruments Inc., Trenton, NJ, USA) mounted on a Zeiss Axiovert 200 microscope with a Plan-Neofluar 40×/0.75 objective with a UV laser. For *A. lixula* eggs with much less transparent cytoplasm than *P. lividus*, egg-incorporated sperm were visualized by incubating the eggs with 25 µM Hoechst-33342 (Sigma–Aldrich) for 5 min before insemination in the fresh seawater. The number of independent experiments (N) and the number of fertilized eggs examined (n) for each condition are shown in Table 1 and Table 2.

### 2.3. Scanning Electron Microscopy (SEM) and Transmission Electron Microscopy (TEM)

For SEM morphological analyses of the egg surface, *P. lividus* eggs were fixed in seawater containing 0.5% glutaraldehyde (pH 8.1) for 1 h at room temperature and post-fixed with 1% osmium tetroxide for an additional hour. Samples were dehydrated in increasing ethanol concentrations, and the subsequent critical point drying was performed with LEICA EM CP300. Specimens were then coated with a thin layer of gold using a LEICA ACE200 sputter coater and observed with a JEOL 6700F scanning electron microscope (Akishima, Tokyo, Japan) or a Zeiss EVO HD 15 scanning electron microscope (Carl Zeiss Microscopy Deutschland GmbH, Oberkochen, Germany). For TEM observations, after the same fixation procedure, eggs were post-fixed with 1% osmium tetroxide and 0.8% K_3_Fe(CN)_6_ for 1 h at 4 °C. After washing in NSW for 10 min twice, the samples were rinsed in distilled water for 10 min twice and finally treated with 0.15% tannic acid for 1 min at room temperature. After extensive rinsing in distilled water (3 times, 10 min each), specimens were dehydrated in increasing ethanol concentrations. Residual ethanol was removed with propylene oxide before embedding in Epon 812. Ultrathin sections were stained with UAR-EMS (Uranyl Acetate Replacement Stain, Electron Microscope Sciences, Hatfield, PA) for 30 min and with 0.3% lead citrate for 30 s before observation with a transmission electron microscope (Zeiss LEO 912 AB, Carl Zeiss Microscopy Deutschland GmbH).

### 2.4. Microinjection, Ca^2+^ Imaging, and Confocal Microscopy

Intact eggs were microinjected using an air pressure transjector (Eppendorf FemtoJet, Hamburg, Germany) as previously described [40]. To monitor Ca^2+^ level increases at fertilization, 500 µM Calcium Green 488 conjugated with 10 kDa dextran were mixed with 35 µM Rhodamine Red (Molecular Probes, Eugene, OR, USA) in the injection buffer (10 mM Hepes, 0.1 M potassium aspartate, pH 7.0) and microinjected into unfertilized eggs. The fluorescence images of cytosolic Ca^2+^ increases were captured with a cooled CCD camera (Micro-Max, Princeton Instruments) mounted on a Zeiss Axiovert 200 microscope with a Plan-Neofluar 40×/0.75 objective at about 3-s intervals, and the data obtained were analyzed with MetaMorph (Universal Imaging Corporation). Following the formula, F_rel_ = [F − F_0_]/F_0_, where F represents the average fluorescence level of the entire egg and F_0_ represents baseline fluorescence, the Ca^2+^ signals detected at a given time point were quantified. Thus, F_rel_ was defined in RFU (relative fluorescence unit) for plotting Ca^2+^ trajectories. The formula F_inst_ = [F_t_–F_(t−1_)]/F_(t−1)_ was applied to analyze the instantaneous increments of the Ca^2+^ level at each moment to visualize the eggs’ local area showing transient Ca^2+^ increase. The changes in the intracellular Ca^2+^ level were analyzed in four independent experiments (N), and the number of eggs (n) being used in each condition is specified in the Results. To visualize F-actin and its remodeling following fertilization, living intact unfertilized eggs were microinjected with 10 µM AlexaFluor568-phalloidin (Molecular Probes, pipette concentration) before insemination and then observed with a Leica TCS SP8X confocal laser scanning microscope equipped with a white light laser and hybrid detectors (Leica Microsystem, Wetzlar, Germany). The number of eggs (n) examined for each condition is reported in the Results.

### 2.5. Statistical Analysis

The numerical MetaMorph data were compiled and analyzed with Excel of Microsoft Office 2010 and reported as “mean ± standard deviation (SD)” in all cases in this manuscript. One-way ANOVA and Mann-Whitney U-test were performed using Prism 5 (GraphPad Software) and at the web site https://www.socscistatistics.com/tests/mannwhitney/default2.aspx, accessed on 11 February 2022), respectively, and *p* < 0.05 was considered statistically significant. For results showing *p* < 0.05, the statistical significance of the difference between the comprising groups was assessed by Tukey post hoc tests.

## 3. Results

### 3.1. Microvillar Reorganization in Unfertilized P. lividus Eggs Exposed to NH_4_0H-Seawater at pH 9

Previous ultrastructural studies using SEM and TEM analyses on several species of unfertilized sea urchin eggs treated with ammonia seawater have shown a gradual elongation of microvilli after 3 h of incubation, with no changes in cortical granules’ morphology [54,57,58]. The present study analyzed differences in the surface topography and the cortical ultrastructure of unfertilized *P. lividus* eggs incubated for 20 min in seawater titrated to pH 9 by NH_4_OH (Figure 1). The control egg showed the typical spherical shape (Figure 1A). By contrast, eggs incubated for 20 min in ammonia seawater (Figure 1B) exhibited an undulating surface due to the structural reorganization of the surface of unfertilized eggs. The short microvilli characteristics of non-activated eggs covering the egg surface are visible at higher magnification. A correlative analysis of the topographical structure of the egg surface (SEM) following alkalinization of the cytoplasm showed that the intracellular pH increase induced alteration of the morphology of microvilli (elongation) in some regions of the egg surface compared to the control (Figure 1C,D). The ultrastructure analysis of the outer cytoplasm of eggs (TEM) evidenced changes in microvilli covered by the vitelline layer and the presence of CG intact beneath the egg plasma membrane in control and eggs incubated in NH_4_OH-seawater at pH 9 (Figure 1E,F). CG in F underwent exocytosis upon insemination.

### 3.2. Eggs Incubated and Fertilized in Ammonia Seawater Display an Altered Ca^2+^ Response at Fertilization

Given that the microvilli on the egg surface are altered by the preincubation of the eggs in ammonia seawater (Figure 1B,D), we examined whether and how the patterns of Ca^2+^ response would change in these eggs at fertilization. Even if 20 min incubation of unfertilized *P. lividus* eggs with ammonia seawater did not induce appreciable Ca^2+^ changes in the resting value of the baseline (data not shown), we found that the Ca^2+^ response at fertilization was altered in a few respects (Figure 2). In sea urchin eggs at fertilization, CF represents the first detectable Ca^2+^ change upon sperm–egg interaction on the egg surface (Figure 2A,B, arrow). CF takes place simultaneously in the cortical region of the egg, and its amplitude is also dependent upon the length and morphology of microvilli [7,17,22,41]. As expected for the unfertilized eggs bearing longer microvilli [22], the amplitude of the CF in the eggs pretreated and fertilized in the ammonia seawater was significantly diminished (0.033 ± 0.01 RFU, *n* = 25) in comparison with the eggs fertilized in NSW (0.078 ± 0.02 RFU, *n* = 32, *p* < 0.01) (Figure 2C,D). The CF precedes the Ca^2+^ wave propagating from the sperm–egg site to the opposite pole (Figure 2A,B, arrowheads). Unlike the CF, the peak amplitude of the Ca^2+^ wave was not significantly changed by the exposure to the ammonia seawater (0.63 ± 0.04 RFU *n* = 25 vs. 0.65 ± 0.07 RFU *n* = 32, Figure 2C). Nonetheless, it is noteworthy that the Ca^2+^ wave in the eggs incubated and fertilized in ammonia seawater experienced a more rapid decline in the intracellular Ca^2+^ level (Figure 2C brown lines, arrowhead) than the control egg (green lines). Whereas the intracellular Ca^2+^ level in the control eggs took 313.6 ± 44.6 s to go back to the basal level (0.2 RFU) after fertilization, the eggs pre-incubated and fertilized in ammonia seawater took only 101.9 ± 14.2 s to arrive at the same level (Figure 2D, *p* < 0.01). The latter result aligns with those obtained from the *P. lividus* eggs incubated and inseminated in seawater titrated to pH 9 by ammonium chloride [22] or with the other weak base, nicotine [42].

### 3.3. Eggs Fertilized in Ammonia Seawater Have a High Tendency of Polyspermy at Fertilization

Given that the microvilli of sea urchin eggs play essential roles in sperm binding and incorporation [7,41,42,59,60], we examined how *P. lividus* unfertilized eggs with altered microvillar structure following ammonia seawater incubation would receive the fertilizing sperm. To this end, the eggs were incubated for 20 min in NSW (pH 8.1) or ammonia seawater (pH 9) and fertilized by Hoechst 33342-stained sperm. Five minutes after insemination, the egg-incorporated sperm were visualized and counted for each egg using epifluorescence microscopy (Figure 3). Although the eggs fertilized in ammonia seawater fully elevated the FE, it turned out that numerous sperm entered the egg (Figure 3B). Indeed, while all of the eggs fertilized in NSW were monospermic (Figure 3A), the average number of sperm inside the eggs fertilized in the ammonia seawater was around 12 (Table 1 and Table 2). It is conceivable that the FE in the eggs inseminated in ammonia seawater may have structural abnormalities related to the overextension of microvilli through the holes on the VL. This might have led to the failure to preclude polyspermic fertilization. On the other hand, similar observations of polyspermy in the eggs exhibiting full elevation of the FE in other experimental conditions argue against the current idea that the primary role of the FE is to mechanically block polyspermy [40,61]. Furthermore, the finding that multiple *Arbacia lixula* and not *P. lividus* sperm can penetrate eggs of the same species following alkalinization of their cytoplasm (data not shown) suggests that even if ammonia seawater enhances the efficiency of the species-specific binding and fusion, the egg’s sperm receptor [10] still discerns acrosome reacted sperm of the same species from those of others.

### 3.4. Effect of Ammonia Seawater on the Cortical F-Actin Changes in Fertilized Eggs

We next applied confocal laser scanning microscopy to visualize F-actin in living eggs microinjected with AlexaFluor568-phalloidin before and after insemination. After 20 min incubation in ammonia seawater, the overall structure of the actin cytoskeleton in the unfertilized egg had not much changed, as judged by the images obtained from the eggs incubated in NSW and ammonia seawater (data not shown). However, the ammonia seawater notably affected the way in which the cortical actin cytoskeleton was reorganized following fertilization (Figure 4). In the eggs fertilized in NSW, the actin filaments near the plasma membrane underwent centripetal translocation with the progress of egg activation (Figure 4A). By contrast, the eggs pre-incubated and fertilized in ammonia seawater failed to translocate the actin fibers efficiently. Most actin fibers did not fully evacuate the outmost cortical region (Figure 4B, *n* = 8). This centripetal F-actin translocation is commonly observed in sea urchin species 10 min after insemination [17,41,42,60,62]. The dramatic reorganization of the cortical F-actin is an indicator of optimum conditions of the eggs showing a normal fertilization response. At variance with the eggs fertilized in NSW, the eggs preincubated and fertilized in ammonia seawater did not display a clear sign of the translocation of the cortical actin filaments as a result of the egg cortex remodeling.

### 3.5. Effects of Ammonia Seawater on the Egg Surface Topography and Cortical Ultrastructure at Fertilization

As mentioned, the egg surface and cortical region undergo drastic rearrangement following fertilization. We investigated whether the preincubation and insemination in ammonia seawater could affect these morphological changes in sea urchin eggs. To this end, we used SEM and TEM to examine the ultrastructure of the eggs’ surface and cortical region, respectively. In Figure 5A,C, scanning electron micrographs showed the surface of a *P. lividus* egg fertilized in NSW (pH 8.1). At 5 min post-fertilization, no sperm were visible on the FE. The higher magnification in Figure 5C highlights the continuous structure of the FE, although the expanded perivitelline space collapsed during the fixation procedure and created a wrinkled appearance. That the egg viewed 5 min after fertilization had undergone extensive cortical granule exocytosis which resulted in the elevation of the FE is verified by the general disappearance of cortical granules in the cortex in the TEM image and by the presence of electron-dense hyaline layer covering the extended microvilli (Figure 5E). The effects of ammonia seawater on the eggs at fertilization may also be found in the structural modification of the VL forming the FE upon insemination. The visible alteration of the FE in these eggs is indicated by: (i) the presence of sperm attached on the FE (Figure 5B,D), implying an increase in the sperm–egg binding capacity, (ii) FE pierced by numerous overextended microvilli (Figure 5D,E), (iii) lack of assembly of the hyaline layer (HL) that was evident in the control egg underneath the FE (Figure 5E,F), (iv) the FE being often interrupted, which appears as a hole in TEM (Figure 5F, arrow). Indeed, microvilli in these eggs are seen to emanate from such openings on the FE. Thus, the exaggerated elongation of microvilli on the surface of eggs treated with ammonia seawater may be in part accountable for the polyspermy, as the over-elongated microvilli may facilitate the fusion and penetration of sperm through the holes of the VL.

### 3.6. Effect of Ammonia Seawater on Embryonic Development

Examination of the early stage (3 h after insemination) of development of the embryos deriving from the eggs incubated and fertilized in NSW pH 8.1 (Figure 6A) or ammonia seawater (Figure 6B) showed abnormal cleavage pattern in the latter embryos, probably due to an excess of paternal chromosomes following polyspermic fertilization caused by the increased receptivity of the eggs to sperm.

## 4. Discussion

The fertilization of starfish and sea urchin eggs is marked by a chain of structural and biochemical changes in the egg cortex resulting in the fusion of the sperm and egg pronuclei. Upon sperm–egg interaction, the generation of the wave-like process of breakdown of the cortical region of the egg cytoplasm (early depolarization phase) is followed by rapid reconstitution (late repolarization phase) [12]. In sea urchin eggs, the early response of the egg to the fertilizing sperm includes depolarization of the membrane potential, which is concurrent with intracellular Ca^2+^ changes and the separation of the VL from the plasma membrane [18,19]. These ionic changes coincide with modifications of the light-scattering properties of the fertilized eggs due to the dehiscence and discharge of cortical granules’ content into the perivitelline space [63,64]. Of the enzymes secreted during the exocytosis of the cortical granules (cortical reaction), one has been suggested to alter the sperm receptor proteins on the VL to preclude the attachment of supernumerary sperm onto the egg coat, which, together with the swelling of the VL, mechanically block polyspermy [65,66,67].

As for the mechanisms leading to the sperm-induced Ca^2+^ signals, recent studies with *P. lividus* suggested that the generation and propagation of the Ca^2+^ influx and Ca^2+^ wave were significantly influenced by the organization of the actin cytoskeleton in the egg cortex and microvilli, as well as by the cortical granules [22,40,41,42]. As shown in Figure 2, a sea urchin egg (*P. lividus*) fertilized in normal conditions (NSW, pH 8.1) readily responds with a cortical flash. The Ca^2+^ wave (arrowhead in A and B), which follows the cortical flash, reaches its peak about one minute after insemination and declines to the baseline approximately 5 min later. During this time of Ca^2+^ signaling, the fertilized egg activates and shows: (i) development of new K^+^ conductance across the cell membrane as a consequence of an increase in the intracellular pH [46,47,68], (ii) stimulation of the metabolic events such as protein and DNA synthesis that are crucial for embryonic development [69]. While early studies highlighted the intracellular Ca^2+^ changes as a primary cause of metabolic activation of the fertilized eggs [70,71], later findings weakened this idea by showing that the incubation of unfertilized sea urchin eggs with ammonia seawater could trigger protein synthesis and chromosome condensation [49,50] in the absence of the early fertilization response, i.e., the Ca^2+^-linked cortical granules exocytosis and separation of the VL. Indeed, it was shown that unfertilized sea urchin eggs exposed to NH_4_OH seawater (pH 9) could mimic what sperm do during the late phases of egg activation by showing: (i) the development of K^+^ conductance, which is characteristic of the late repolarization seen in the fertilization process, (ii) increase in intracellular pH, (iii) increased rate of protein synthesis. All of these effects were observed without the intracellular Ca^2+^-linked cortical granules exocytosis. These findings added weight to the suggestion that the first early phase was not a prerequisite for the late metabolic activation of the egg [48,54].

From a structural point of view, it is well known that the late phase of the fertilization process of sea urchin eggs (phase III) [48] starting 5 min after insemination is coincident with the decay of the Ca^2+^ wave down to the baseline levels (Figure 2C) and is marked by a dramatic restructuration of the egg cortex. This includes dynamic structural changes in actin in the egg cortex and microvilli elongation in the perivitelline space due to increased intracellular pH [43,44,55,72]. Microvilli elongation resulting from actin polymerization, which necessitates the uptake of external Ca^2+^, can also be induced in unfertilized sea urchin eggs following their incubation in 10 mM NH_4_Cl seawater for several hours [57]. Similarly, the incubation of sea urchin sperm in ammonia seawater at pH 9 for 10 min promotes the Ca^2+^ uptake required for actin polymerization to form acrosomal filament in the absence of the jelly coat, the natural stimulus triggering the acrosomal reaction [73,74]. Indeed, Ca^2+^ uptake and pH increase are essential for inducing the formation of the long actin filament of the acrosomal process in starfish [75,76]. As for the molecular mechanism leading to the formation of the acrosomal process, it has been suggested that the intracellular pH would induce the dissociation of inhibitory proteins from the actin, allowing it to undergo polymerization [76].

In the present contribution, the results of the sperm-induced Ca^2+^ signals in *P. lividus* (control and ammonia-seawater treated eggs) have confirmed our previous findings on the relationship between microvilli morphology and the amplitude of CF in fertilized sea urchin eggs [17,22,41,42,60]. As evidenced in the histograms of Figure 2C, the CF amplitude was significantly reduced when *P. lividus* eggs were pre-incubated and inseminated in ammonia seawater. In this respect, the elongation of microvilli seen in the SEM and TEM images of the unfertilized eggs pre-incubated in ammonia seawater (Figure 1) may have hampered microvilli reorganization occurring upon sperm stimulation which is necessary to promote a normal Ca^2+^ influx during CF [17,22,41,42,77,78,79].

It is noteworthy that ammonia seawater induced microvilli elongation without affecting the structure of the egg cortex and cortical granules, which underwent normal exocytosis upon insemination (Figure 1F and Figure 5F), as previously reported in another species of sea urchin [54]. The selective effect of ammonia seawater on microvillar morphology lends credence to the idea that the modulation of CF amplitude is attributable to the F-actin-based structure of microvilli. In line with this, other amines, such as urethane and procaine, affecting their morphology altered the CF amplitude [22].

Another conspicuous finding in this work is that eggs pre-incubated and fertilized in ammonia seawater displayed a faster decline in the intracellular Ca^2+^ level down to the baseline levels (Figure 2C). Similarly, the rapid decline in the sperm-induced Ca^2+^ signals has also been commonly observed in *P. lividus* eggs treated and fertilized in seawater containing other weak bases and alkaloid amines, e.g., NH_4_Cl, urethane, procaine, Gly-Phe-*β*-naphthylamide (GPN), and nicotine [22,42]. While these membrane-permeant agents may produce a faster decline in intracellular Ca^2+^ level based on a common pathway such as an increase in intracellular pH, the exact mechanism is not known. Nonetheless, a few scenarios are conceivable. First, the function of the Ca^2+^ reabsorption system in the egg that usually removes cytosolic Ca^2+^ may have been enhanced in the given intracellular condition, such as cytoplasmic alkalinity. Here, Ca^2+^ uptake in acidic cortical organelles such as cortical granules or the endoplasmic reticulum (ER) associated with them are possible candidates [22,80,81,82]. However, cortical granules in ammonia-treated eggs underwent exocytosis upon insemination, resulting in the elevation of the FE (Figure 3 and Figure 5), and data on physiological intracellular alkaline shifts have shown cytosolic Ca^2+^ increases due to the depletion of the ER Ca^2+^ store [83,84]. Another possibility is that the hyperpolymerization of microvilli following the intracellular Ca^2+^ increase may have absorbed free Ca^2+^ into the growing actin filaments [57]. Indeed, SEM and TEM images (Figure 5) show that 5 min after insemination, ammonia seawater induced an exaggerated elongation of microvilli in the perivitelline space. As a Ca^2+^-binding protein with extremely high affinity, actin monomers can readily sequester Ca^2+^ when its concentration is raised and polymerize themselves into F-actin, in which Ca^2+^ is no longer accessible for exchange. In this way, polymerizing actin filaments can serve as a Ca^2+^ reservoir [77,78,85]. Thus, the formation of much longer microvilli following alkalinization of the egg cytoplasm may explain the faster decay time of the Ca^2+^ wave to reach 0.2 RFU of Ca^2+^ levels.

Microvilli elongation induced by internal pH increase [57,86] and the changes in K^+^ conductance in excitable cells [48,87,88] support the hypothesis that some specific ion conduction may be modulated by microvillar actin cytoskeleton [77,78]. In line with this, immature starfish oocytes, usually prone to polyspermy [61,89,90,91], undergo a decline in K^+^ conductance during the maturation process. In this period of meiotic maturation, the microvilli of the maturing oocyte become shorter [7,38,92]. Indeed, proper structural organization of microvilli and cortical granules beneath the plasma membrane of mature eggs ensures the generation of a normal fertilization potential, Ca^2+^ response, cortical reaction, and monospermic fertilization [7,38,92,93,94]. The results of this contribution support the idea that the organization of microvilli has a profound effect on the activation of the Ca^2+^ influx and the regulation of the pattern of the Ca^2+^ wave (decay). Furthermore, when depolymerization of cortical actin filaments is induced by latrunculin in starfish eggs (*Astropecten aranciacus*), the VL separates from the plasma membrane following a Ca^2+^ increase, and the activated eggs exhibit durable membrane depolarization in a Ca^2+^- and Na^+^-dependent manner, which is reminiscent of what is observed in fertilized eggs [45,95,96]. Furthermore, pre-treatment of unfertilized *P. lividus* eggs with the actin drugs generates an altered CF and Ca^2+^ wave [22,40]. Thus, whatever ion channels are involved in these ion flux events, they are heavily influenced by the F-actin-based structural organization of microvilli and cortical granules tightly associated with them to the extent that a normal Ca^2+^ response is triggered upon sperm stimulation.

Although the effect of weak and strong bases in inducing artificial parthenogenesis has been studied for more than a hundred years [97], our results have shown that, in addition to an altered Ca^2+^ response, the preincubation and fertilization of the eggs in ammonia seawater give rise to polyspermy upon insemination [58]. The cause of polyspermy lies in the chemical nature of ammonia, which efficiently increases intracellular pH and thereby affects a variety of proteins sensitive to that pH shift. This idea is supported by the finding that seawater with the same alkalinity (pH 9.0) adjusted with strong bases NaOH (*n* = 60) or KOH (*n* = 20) had no such effect; all fertilized eggs were monospermic because strong bases are much less membrane permeant [48]. Hence, it is not likely that the cell surface proteins such as sperm receptors became hyper-activated by alkalinity itself to capture more sperm and internalize them. It is more likely that the effect of polyspermy is attributed to the membrane-permeant capability of the dissociated form of ammonia. Once inside the cell, it is expected to receive a proton and increase intracellular pH as a base [98,99]. Thus, the polyspermy here may be primarily attributable to some event inside the egg cytoplasm whose effect is extended to the cell surface. In particular, the hyperreactive extension of microvilli at the surface of unfertilized eggs may be relevant to the overwhelming tendency of polyspermy.

The polyspermic response of *P. lividus* eggs fertilized in ammonia seawater is similar to that of the eggs fertilized in the presence of nicotine in the sense that actin filaments engulfing multiple sperm at that egg surface are often hyperactive and in favor to form thick F-actin bundles extending well into and often beyond the perivitelline space [42]. In the sea urchin eggs exposed to ammonia seawater, the overextension of microvilli was so evident before and after fertilization to the extent that the tip of the outgrowing microvilli often went out of the perivitelline space through the apparent “holes” on the FE (Figure 1 and Figure 5). While the pattern of polyspermy displayed by *P. lividus* eggs fertilized in ammonia seawater (Table 1 and Table 2) is very similar to that of the eggs pre-incubated and fertilized in the presence of nicotine, e.g., the number of egg-incorporated sperm and the frequency of polyspermic fertilization [42], it is less likely that the polyspermy in nicotine-exposed eggs is mainly due to the increase in intracellular pH potentially caused by nicotine. Firstly, if the change in cytoplasmic alkalinity itself is the only way that nicotine gives rise to polyspermy, the stereoisomers of nicotine are expected to have virtually the same tendency in causing polyspermy. However, (−) nicotine was two times more effective than (±) nicotine in inducing polyspermy in sea urchin eggs [42]. Secondly, nicotine is a weaker base (alkaloid), with its second pKa value at 7.9, whereas pKa of ammonia is 9.36. Thus, nicotine is much less effective as a base in the egg’s cytosolic pH, which is estimated to be around 6.84 in the case of *Lytechinus pictus* [47]. However, it was nicotine, and not ammonia (data not shown), that was able to drastically change the structure of the cortical actin cytoskeleton in the unfertilized eggs of *P. lividus* [42]. Thus, the way nicotine affects the actin cytoskeleton appears to still be different from that of ammonia seawater (see also the apparent lack of cortical granules exocytosis as judged by light microscopy observations). Since ammonia seawater elongated the microvilli on the surface of unfertilized eggs, which extended even beyond the FE following fertilization, indeed, it is this overextension of microvilli that may be related to polyspermy in these eggs. Although *P. lividus* eggs pre-incubated and fertilized in the presence of 10 mM dithiothreitol (DTT) exhibited not only deregulated centripetal translocation of F-actin (similar to what is shown in Figure 4) but also prominent elongation of microvilli following fertilization [60], they were consistently monospermic upon insemination. It should also be noted that 20 min incubation of sea urchin (*Lytechinus pictus*) eggs in ammonia sewater shifts the eggs’ membrane potential to be more negative (by >30 mV) and that the plasma membrane develops K^+^ permeability [48]. This profound change in the electrical property of the plasma membrane may make a difference to the sperm’s efficiency in generating the egg’s fertilization potential and its pattern, which was suggested as an underlying mechanism for the fast block to polyspermy. However, according to the same authors [48], the eggs exposed to ammonia seawater (and thereby attaining hyperpolarized membrane potential) exhibited normal fertilization potential, arriving at about +10 mV, which is supposed to be high enough to block polyspermy in the electrical mechanism [16]. While the latter point awaits careful experimental investigation in *P. lividus* eggs in the given conditions, the simple finding that polyspermy in ammonia seawater takes place despite the elevation of FE suggests that the structural alteration of the FE is insufficient to block polyspermy.

## 5. Conclusions

Sea urchin eggs respond to the fertilizing sperm with an intracellular Ca^2+^ increase and membrane separation from the egg surface within 1 min after fertilization, a time during which the susceptibility of the eggs is very high due to the disintegration of the egg cortex [100]. This early phase is followed by a series of late changes beginning about 5 min after sperm–egg contact, including the cortical structural and metabolic changes necessary for subsequent embryo development. Our previous studies have highlighted the importance of egg quality characterized by optimal actin-linked structural organization of the egg cortex that enables the egg to successfully respond to the fertilizing sperm with a normal Ca^2+^ response concomitant with cortical granule exocytosis. This early phase of egg activation is a sine qua non condition for regulating monospermic incorporation as a result of the dynamic cortical cytoskeletal rearrangement and for the success of embryo development [15,101]. Our present study has suggested the strict interdependence between the intracellular pH increase occurring during the late phase of the fertilization response and the regulation of intracellular Ca^2+^ levels following the F-actin-based structural remodeling of the cortex of fertilized eggs. Even if biochemical mechanisms underlying the relationship between protons and Ca^2+^ concentration in cell physiology to regulate various channels, pumps, exchangers, organelles, and metabolic events are well established [102,103,104], the data in the literature on the correlation between intracellular pH and cytosolic levels of Ca^2+^ are scarce and therefore merit further investigation in the future.

## Figures and Tables

**Figure 1 cells-11-01496-f001:**
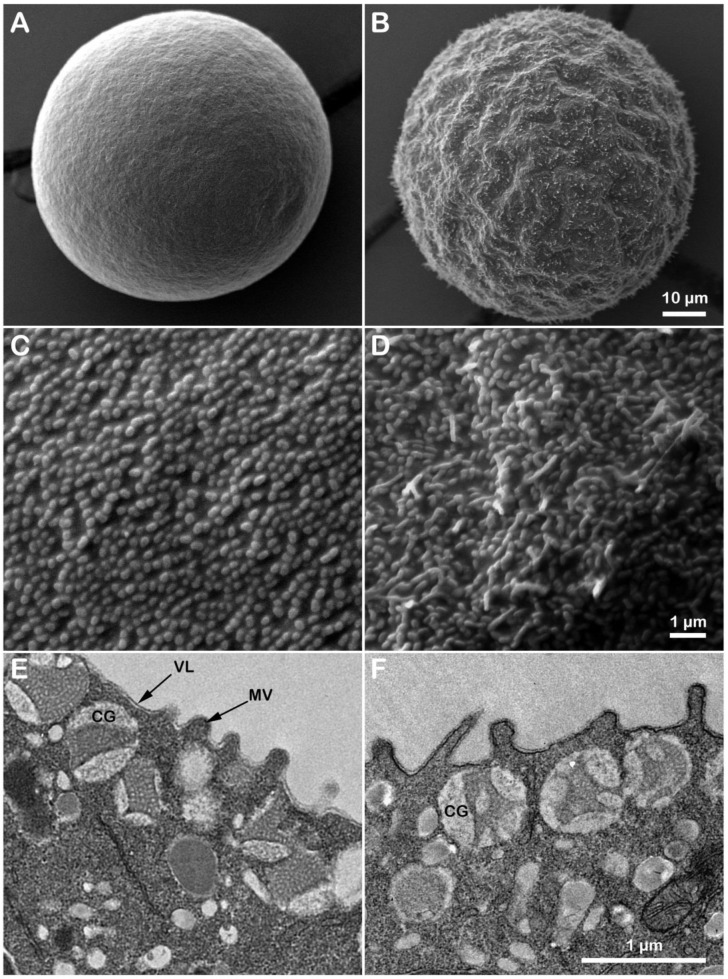
Effect of ammonia seawater on the outer surface of unfertilized *P. lividus* eggs. Before fixation, *P. lividus* eggs of the same batch were partitioned and incubated for 20 min either in NSW at pH 8.1 (control) or in SW adjusted to pH 9 using NH_4_OH (ammonia seawater). Changes in the outer egg surface and cortical ultrastructure were observed by SEM (**A**–**D**) or TEM (**E**,**F**). Note the eggs’ undulated surface (**B**) and microvilli’s changes in shape and length (**D**). Microvilli covered by the vitelline layer are relatively uniform on the surface of the control egg (**C**,**E**) but are often elongated on the surface of eggs incubated in ammonia seawater (**D**,**F**). Abbreviations: VL, vitelline layer; MV, microvilli; CG, cortical granules.

**Figure 2 cells-11-01496-f002:**
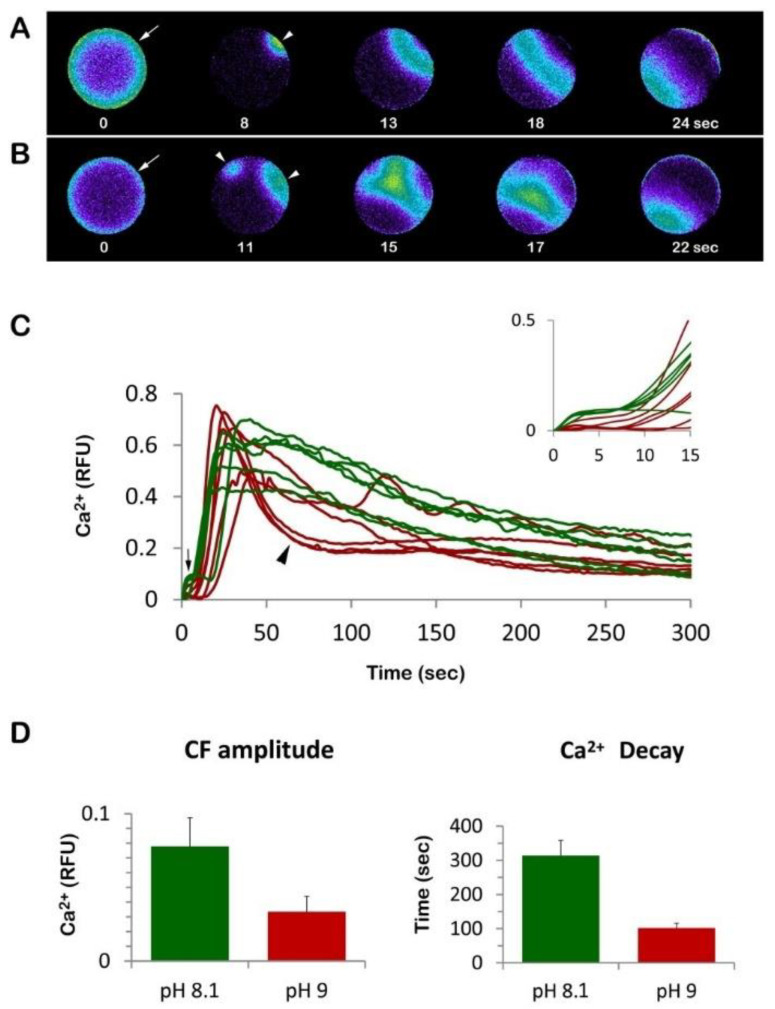
Sea urchin eggs fertilized in ammonia seawater display altered Ca^2+^ response. P. lividus eggs were microinjected with the calcium dye and incubated for 20 min either in NSW pH 8.1. (control) or in ammonia seawater. Then, the sperm-induced Ca^2+^ increases in these eggs were compared. (**A**) Momentary increases in Ca^2+^ levels were visualized in pseudocolor in the control egg (fertilized in NSW) and in the egg fertilized in ammonia seawater (**B**). The time point showing cortical flash (CF, indicated by an arrow) was set as *t* = 0. The arrowheads indicate the sperm interaction sites on the egg surface and the Ca^2+^ waves triggered by two sperm. (**C**) The trajectory of the intracellular Ca^2+^ levels in control eggs (green curves) and the eggs fertilized in ammonia seawater (brown curves). The vertical arrow indicates the CF, and the arrowhead indicates the more abruptly decaying intracellular Ca^2+^ levels 300 s after insemination. The inset shows the same Ca^2+^ trajectories plotted on a different time scale to visualize the initial Ca^2+^ changes more clearly. (**D**) Histograms showing the most affected aspects of Ca^2+^ signal at fertilization: amplitude of the cortical flash (CF) and declining kinetics of the Ca^2+^ wave expressed in terms of the time required for the Ca^2+^ level RFU to reach 0.2 RFU. Tukey HSD test, *p* < 0.01. Abbreviation: RFU, relative fluorescence unit.

**Figure 3 cells-11-01496-f003:**
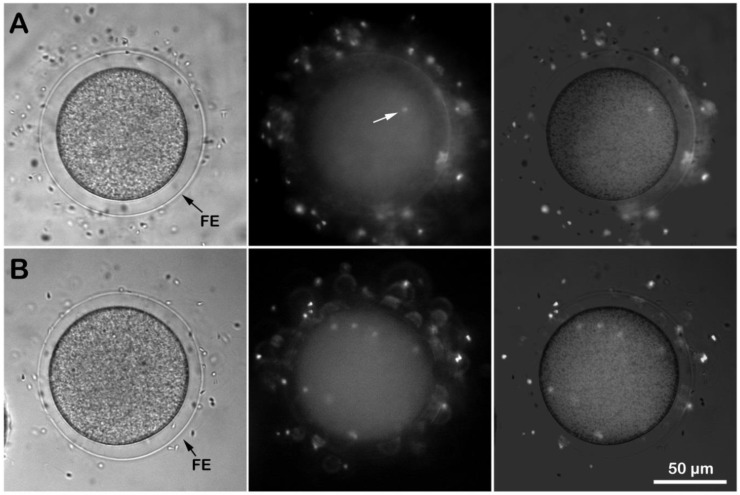
Effect of ammonia seawater on the egg receptivity to sperm. *P. lividus* eggs pre-incubated for 20 min in NSW (pH 8.1) (**A**) or in ammonia seawater (pH 9) were fertilized by Hoechst 33342-stained sperm (**B**). The number of sperm in the living zygote was counted 5 min after insemination by a CCD camera with a UV laser in epifluorescence microscopy (middle panel). The elevation of the fertilization envelope (FE) was visualized in the bright field view, and the merged image was provided to distinguish sperm inside the egg from those attached to the FE.

**Figure 4 cells-11-01496-f004:**
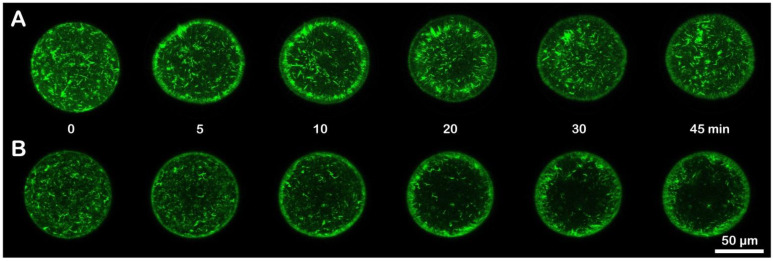
Effect of ammonia seawater on F-actin dynamics in fertilized eggs. *P. lividus* eggs were microinjected with AlexaFluor568-phalloidin and incubated for 20 min in normal seawater (NSW, pH 8.1) (**A**) or in ammonia seawater (pH 9) prior to fertilization in the same medium (**B**). The structural changes in the actin cytoskeleton following fertilization were monitored with the laser scanning confocal microscope. The moment of sperm addition was set as *t* = 0.

**Figure 5 cells-11-01496-f005:**
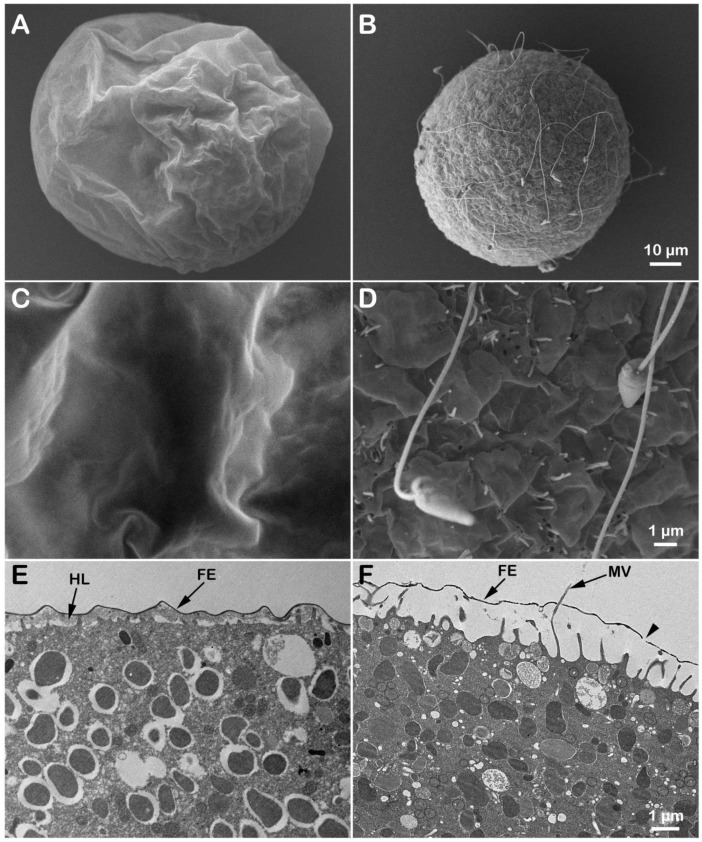
Surface topography and cortical ultrastructure of the eggs fertilized in ammonia seawater. *P. lividus* eggs were fertilized and fixed 5 min after insemination for SEM (**A**–**D**) and TEM (**E**,**F**) observations. (**A**,**C**) Control eggs fertilized in NSW at pH 8.1. (**B**,**D**) Eggs were pre-incubated (20 min) and fertilized in ammonia seawater at pH 9. Enlarged view of the elevated FE covering the control egg (**C**) and the punctured FE of the egg fertilized in ammonia seawater (**D**). Note in panel D the over-elongated microvilli passing through the punctured FE. Abbreviations: FE, fertilization envelope; HL, hyaline layer; MV, microvilli.

**Figure 6 cells-11-01496-f006:**
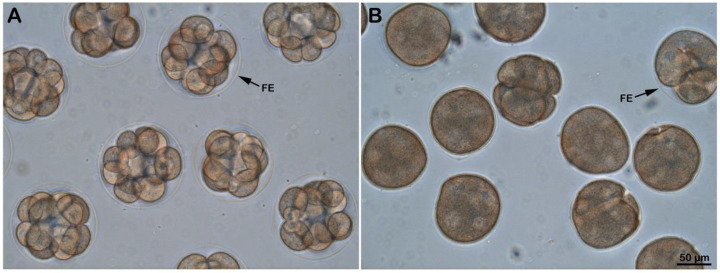
Abnormal development of the eggs fertilized following alkalinization of their cytoplasm. *P. lividus* eggs were fertilized after incubation in normal seawater (NSW, pH 8.1) (**A**) or ammonia seawater (pH 9) (**B**), and then the subsequent development was monitored by light microscopy after 3 hr. Note that the FE, which had undergone elevation in the eggs incubated and fertilized in ammonia seawater, is seen collapsed on the surface of embryos after 3 h of insemination.

**Table 1 cells-11-01496-t001:** Frequency of Polyspermy (%) in *P. lividus* and *A. lixula* eggs Fertilized in NSW (pH 8.1) and Ammonia SW (pH 9).

	*P. lividus* NSW	*P. lividus* pH 9	*A. lixula* NSW	*A. lixula* pH 9
**Mean**	0	100 *	5	100
**SD**	0	0	0.01	0
** *n* **	4	4	2	2

Note: * *p* < 0.01 in Tukey test compared to fertilization in NSW for the corresponding species.

**Table 2 cells-11-01496-t002:** Number of Sperm Inside *P. lividus* and *A. lixula* Eggs Fertilized in NSW (pH 8.1) and Ammonia SW (pH 9).

	*P. lividus* NSW	*P. lividus* pH 9	*A. lixula* NSW	*A. lixula* pH 9
**Mean**	1	12.89 *	1.07	6.07 *
**SD**	0	5.79	0.35	1.33
** *n* **	80	80	40	40

Note: * *p* < 0.00001 in U-test compared to the control (fertilization in natural seawater, NSW for the corresponding species).

## Data Availability

Not applicable.

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
