# Peer review of "Regulation of the Actin Cytoskeleton-Linked Ca2+ Signaling by Intracellular pH in Fertilized Eggs of Sea Urchin"

_cells, 2022, doi:10.3390/cells11091496_

Round 1

Reviewer 1 Report

The manuscript entitled “Regulation of the actin cytoskeleton-linked Ca2+ signaling by intracellular pH in fertilized eggs of sea urchin”by LImatola et al. examined the effect of the higher pHi in egg cells on fertilization using sea urchin eggs. Although this is an artificial experimental system, the results are interesting because they show that high pH is the cause of polyspermy.

On the other hand, I think that the sea urchin eggs show increase in intracellular pH after fertilization, and that the forced pHi increase is expected to mimic the post-fertilization state. However, this is not the case in the results of this study: it seems to prevent the post-fertilization state, such as block of polyspermy. The manuscript attributes all the effects of high pH to the length of microvilli, but this seems a weak explanation for the inability to mimic the post-fertilization state. Also, why does the pH increase cause disorder of actin filaments, even though the change is also seen in normal cells? I think it is necessary to discuss the point in more depth, including additional experiments.

In addition, there are other problems listed below that need to be corrected.

Page 5 lines 213-215, and Figure. 1:
In the results, the authors stated that “TEM images corroborated the overextension of microvilli underneath the VL of the eggs fertilized in ammonia seawater compared with the microvilli in control eggs (Fig. 1E and F)”, but it is not obvious that microvilli shown in Fig. 1F is longer than that shown in Fig. 1E.  The photo of Fig. 1F should be replaced with a clearer one or the expression of the result should be weakened.

Page 8, Part 3.3 and Figure 3:
An epifluorescence microscope was used for observation in this experiment. However, such a microscope should not be able to strictly distinguish whether the sperm enter the egg or are attached on the egg surface, because of the depth of focus.

On the other hand, if there are no sperm between FE and the egg plasma membrane even in the ammonia-seawater-treated eggs, FE certainly acts as a physical barrier to sperm, and "these results argue against the long-standing idea that the Is not the "primary role of the FE is to mechanically block the entry of supernumerary sperm." (lines 277-278) is an exaggeration.

Page 9-10, Part 3.5 and Figure 5:
In relation to the above comment, it is very interesting that there are microvilli penetrating the FE in the ammonia-seawater-treated eggs. It would be interesting to know if the polyspermy is caused by the entry of sperm through the holes on the FE or if there is some interaction between the microvilli and sperm outside the FE.

Page 13 lines 427-430:
The relationship between cortical flash and length of microvilli is parallel, but it is not clear here whether there is any correlation. The expression should be weakened.

Minor point

”Fig.” and ”Figure” are intermingled as expressions in the main text. Either of them should be unified.

Author Response

The manuscript entitled "Regulation of the actin cytoskeleton-linked Ca2+ signaling by intracellular pH in fertilized eggs of sea urchin" by LImatola et al. examined the effect of the higher pHi in egg cells on fertilization using sea urchin eggs. Although this is an artificial experimental system, the results are interesting because they show that high pH is the cause of polyspermy.

On the other hand, I think that the sea urchin eggs show increase in intracellular pH after fertilization, and that the forced pHi increase is expected to mimic the post-fertilization state. However, this is not the case in the results of this study: it seems to prevent the post-fertilization state, such as block of polyspermy. The manuscript attributes all the effects of high pH to the length of microvilli, but this seems a weak explanation for the inability to mimic the post-fertilization state. Also, why does the pH increase cause disorder of actin filaments, even though the change is also seen in normal cells? I think it is necessary to discuss the point in more depth, including additional experiments.

We thank the reviewer for judging our results interesting. As animal models, sea urchins and other marine invertebrates have profoundly contributed for more than 100 years to our understanding of oocyte maturation, fertilization, and embryonic development due to the availability of a large number of gametes and the possibility of performing in vitro studies in the same conditions as in nature, i.e., seawater. In particular, our findings on the importance of the structure and dynamics of the actin cytoskeleton in starfish eggs have suggested further investigative avenues to preserve actin cytoskeletal dynamics at fertilization as a part of human-assisted oocyte activation (AOA) (Kashir et al. 2022 Human Reproduction Open p.18). 

We also thank the reviewer for pointing out that the forced pHi increase is expected to mimic the post-fertilization state but caused polyspermy, which gives us the chance to clarify the point. Our results suggested that pre-alkalinization of the cytoplasm (pH increase by ammonia) of unfertilized eggs that induced polyspermy might be related to the elongation of microvilli. Indeed, it has been reported that microvilli are involved in sperm fusion (Developmental Biology 9th edition, Scott. F. Gilbert). Furthermore, the alteration of the structure of the vitelline layer, which is only visible after the elevation of the FE in eggs fixed 5 minutes after insemination (Fig. 5 F), permitting the exposure of microvilli through the holes on the vitelline layer is also compatible with the idea. The block to polyspermy is established a few seconds after insemination. We have previously shown that it is mediated by structural and not electrical changes in the egg plasma membrane (Limatola et al. 2019, Zygote, ref. 17). In this sense, it is likely that pre-alkalization of the egg has changed how the egg interacts and internalizes sperm to bring about monospermic fertilization. As discussed in the text, it is not likely that the observed polyspermy is because the ammonia-pretreated eggs failed to produce positive-going fertilization potential. Indeed, sea urchin eggs in the same condition produced nearly indistinguishable fertilization potential (Steinhardt and Mazia 1973, Ref. 48). With the possibility of the electrical block to polyspermy being ruled out or weakened, the most plausible explanation for our observation is a mechanism based on the actin filaments in the microvilli and egg surface.

Our study suggests that by altering the structural organization of the actin cytoskeleton of the egg cortex of unfertilized eggs, the pH increase before insemination affects the later F-actin translocation in fertilized sea urchin eggs (Fig 4), which normally occurs in the late phase of the fertilization process [see ref. 62] and 4 of our published papers [17, 41,42,60]. The remaining question is about the mechanism by which the pHi shift causes such changes in the structure and dynamics of the actin cytoskeleton. This is an interesting question, but answering it in individual molecules has not been much addressed. Obviously, pH can affect the activities of several actin-binding proteins such as cofilin, gelsolin, and formin. Still, the question of the detailed molecular mechanisms concertedly producing the effect is beyond the scope of our present study and deserves a formally dedicated investigation in the future.

In addition, there are other problems listed below that need to be corrected.

Page 5 lines 213-215, and Figure. 1:

In the results, the authors stated that "TEM images corroborated the overextension of microvilli underneath the VL of the eggs fertilized in ammonia seawater compared with the microvilli in control eggs (Fig. 1E and F)", but it is not obvious that microvilli shown in Fig. 1F is longer than that shown in Fig. 1E. The photo of Fig. 1F should be replaced with a clearer one or the expression of the result should be weakened.

We showed the results of scanning electron microscopy (SEM) and transmission electron microscopy (TEM) analyses to visualize the topography of the egg surface and the ultrastructure of the egg cortex. The alkalinization of the cytoplasm of unfertilized eggs induced changes in microvillar morphology, and this is well evident in the SEM and TEM micrographs at lower and higher magnifications. The problem with TEM observations is that it is not possible to visualize the complete length of microvilli due to the sectioning of the samples, which may cut their tips. In other words, longitudinal sectioning of an entire microvillus is difficult to obtain. Furthermore, elongation of microvilli is not uniform all over the egg surface. We have modified the sentence in "A correlative analysis at a higher magnification of the topographical structure of the egg surface (D), and ultrastructure of the outer cytoplasm of the eggs (F) showed that the alkalinization of the cytoplasm of unfertilized eggs induced alteration of the morphology of microvilli (elongation) in some regions of the egg surface. Note the presence of CG intact beneath the egg plasma membrane covered by the vitelline layer following cytoplasmic alkalinization (Fig. 1F), which underwent exocytosis upon insemination (Fig. 5 F)".

Page 8, Part 3.3 and Figure 3:

An epifluorescence microscope was used for observation in this experiment. However, such a microscope should not be able to strictly distinguish whether the sperm enter the egg or are attached on the egg surface, because of the depth of focus.

On the other hand, if there are no sperm between FE and the egg plasma membrane even in the ammonia-seawater-treated eggs, FE certainly acts as a physical barrier to sperm, and "these results argue against the long-standing idea that the Is not the "primary role of the FE is to mechanically block the entry of supernumerary sperm." (lines 277-278) is an exaggeration.

We understand the concern expressed by the reviewer; however, we would like to reassure the reviewer that our methodology can clearly distinguish whether one or multiple spermatozoa are inside or outside living fertilized eggs. This is so because 5 minutes after insemination, the DNA of the sperm in the egg undergoes decondensation, and it is thus easy to distinguish between spermatozoa incorporated into eggs from those outside. Among others, we have published 8 papers using such a sperm count methodology.

Our statement underlines that the FE is not there to prevent polyspermy. In line with this, our previously published results have shown, at variance with the prevailing view, that in starfish and sea urchins, the fertilization envelope (FE) does not block the entry of supernumerary sperm. We have suggested and shown that immediately after the attachment of the fertilizing sperm to the egg surface, rapid structural changes and not electrical events may prevent the binding and fusion of additional sperm. So sperm cannot be seen in the perivitelline space because they are not bound to the egg surface even before the elevation of the FE (see Fig. 1 in Limatola et al. 2019 Zygote, 27: 241-249, "Sodium-mediated fast electrical depolarization does not prevent polyspermic fertilization in Paracentrotus lividus eggs" https://doi.org/10.1017/S0967199419000364, Ref 17). We have added the following sentence in the Introduction, line 60: The results also showed that a structural block to polyspermy is established by the attachment of the fertilizing sperm to the egg surface, which prevents the binding and fusion of additional sperm before the separation of the vitelline layer [14,17].

Page 9-10, Part 3.5 and Figure 5:

In relation to the above comment, it is very interesting that there are microvilli penetrating the FE in the ammonia-seawater-treated eggs. It would be interesting to know if the polyspermy is caused by the entry of sperm through the holes on the FE or if there is some interaction between the microvilli and sperm outside the FE.

That the structure of the vitelline layer (precursor of the fertilization envelope, FE) has been altered by the alkalinization of the cytoplasm of unfertilized eggs is evident only after fertilization. Indeed the effect of intracytoplasmic pH increase in addition to the microvillar elongation in unfertilized eggs induced an exaggerated elongation of microvilli also in the post-fertilization phase (4-5 minutes after insemination) that protrude from the holes of the FE. This is what we have written in the discussion: Thus, the elongation of microvilli on the surface of unfertilized eggs treated with ammonia seawater may be in part accountable for the polyspermy, as it may facilitate fusion and penetration of sperm through the holes of the VL. In other words, only after fertilization is it possible to visualize ruptures at the level of the vitelline layer, which will form the FE. We have thus suggested that polyspermy may occur because the holes of the vitelline layer (precursor of the FE) allow multiple sperm to bind and fuse with longer microvilli at the sites where the vitelline layer is discontinuous.

Page 13 lines 427-430:

The relationship between cortical flash and length of microvilli is parallel, but it is not clear here whether there is any correlation. The expression should be weakened.

We thank the reviewer for allowing us to quote our published papers, namely [17,22,41,42,93] on the relationship between microvilli morphology and cortical flash and not only the papers by Lange (1999, 2000), who pioneered the idea of the role of microvilli in Ca2+ signaling.

Minor point

" Fig." and" Figure" are intermingled as expressions in the main text. Either of them should be unified.

We have unified the intermingled expressions of "Fig." and "Figure" as requested.

Reviewer 2 Report

minor issues

I was confused in reading Table 1&2: 

Table 1 & 2 show that the % of oocytes exhibiting polyspermy at elevated pH was higher than at normal pH in both P. lividus & A. Lixula.  Also, the number of incorporated sperm in eggs fertilized at high pH was greater than in eggs fertilized at the normal pH in both P. lividus & A. Lixula.  This seems to contradict the statement (line 279-282) ‘multiple Arbacia lixula and not P. lividus sperm can penetrate eggs of the same species’  because it looks like these tables did not include cross fertilization experiments.

Line 380 and 381 don't seem to fit

Line 495 should be "receive a proton"

Linse 541 should read "insufficient to block polyspermy"

Author Response

minor issues

I was confused in reading Table 1&2:

Table 1 & 2 show that the % of oocytes exhibiting polyspermy at elevated pH was higher than at normal pH in both P. lividus & A. Lixula. Also, the number of incorporated sperm in eggs fertilized at high pH was greater than in eggs fertilized at the normal pH in both P. lividus & A. Lixula.  This seems to contradict the statement (line 279-282) 'multiple Arbacia lixula and not P. lividus sperm can penetrate eggs of the same species' because it looks like these tables did not include cross fertilization experiments.

We thank the reviewer for his positive evaluation of the results of our study and for asking to clarify the point concerning the cross-fertilization data missing in Tables 1 and 2. The reviewer was right about the confusion originating from the unclear sentence. We decided not to insert the cross-fertilization data because alkalinization of the cytoplasm of P. lividus eggs did not promote heterologous fertilization with A. lixula sperm. We only discussed this result. We have modified the sentence as follows:

Furthermore, the finding that multiple A. lixula and not P. lividus sperm can penetrate eggs only of the same species following alkalinization of their cytoplasm (data not shown) suggests that even if ammonia seawater may have enhanced the efficiency of the species-specific binding and fusion, the egg's sperm receptor [10] still discern acrosome reacted sperm of the same species from those of others.

Line 380 and 381 don't seem to fit

We apologize for having inserted a sentence that does not fit with the discussion content by mistake. We have eliminated it.

Line 495 should be "receive a proton"

We have replaced ….to receive proton… with to receive a proton…. as suggested by the reviewer.

Linse 541 should read "insufficient to block polyspermy"

We have replaced …is insufficient to work to block polyspermy with …. is insufficient to block polyspermy as suggested by the reviewer.

Round 2

Reviewer 1 Report

Basically, I acknowledge the revised manuscript. I still have the opinion that Fig. 1F should be replaced, but for the moment, I am okay with this revision of the description.

However, the still existing claim "These results argue against the current idea that the primary role of the FE is to block the entry of supernumerary sperm mechanically.” (Page 9, lines 294-295) is unacceptable. 

Even if the fact that "sperms cannot be seen in the perivitelline space because they are not bound to the egg surface even before the elevation of the FE" as stated by the authors’ response is the case, it does not mean that FE is meaningless in the polyspermy block. Could the reason for polyspermy in the pre-alkalinized eggs be that the FE had  holes on it or that the microvilli had extended through the FE so that the FE no longer acted as a barrier for the polyspermy block?

Author Response

We thank the reviewer for having acknowledged the first revision of our manuscript. In the requested second revision, we have accepted his suggestion that the alteration of the structure of the VL, the precursor of the fertilization envelope (FE), following alkalinization of the egg cytoplasm may have failed to block polyspermy.  Thus we have modified the sentence on page 9, line 294-295, as follow:

It is conceivable that the FE in the eggs inseminated in ammonia seawater may have structural abnormalities related to the overextension of microvilli through the holes on the VL. This might have led to the failure to preclude polyspermic fertilization. On the other hand, similar observations of polyspermy in the eggs exhibiting full elevation of the FE in different experimental conditions argue against the current idea that the primary role of the FE is to mechanically block polyspermy.